# Spinochrome D Attenuates Doxorubicin-Induced Cardiomyocyte Death via Improving Glutathione Metabolism and Attenuating Oxidative Stress

**DOI:** 10.3390/md17010002

**Published:** 2018-12-20

**Authors:** Chang Shin Yoon, Hyoung Kyu Kim, Natalia P. Mishchenko, Elena A. Vasileva, Sergey A. Fedoreyev, Valentin A. Stonik, Jin Han

**Affiliations:** 1National Research Laboratory for Mitochondrial Signaling, Department of Physiology, College of Medicine, Cardiovascular and Metabolic Disease Center (CMDC), Inje University, Busan 614-735, Korea; changshin73@gmail.com (C.S.Y.); estrus74@gmail.com (H.K.K.); 2G.B. Elyakov Pacific Institute of Bioorganic Chemistry, Far-Eastern Branch of the Russian Academy of Science, Vladivostok 690022, Russia; mischenkonp@mail.ru (N.P.M.); vasilieva_el_an@mail.ru (E.A.V.); fedoreev-s@mail.ru (S.A.F.); stonik@piboc.dvo.ru (V.A.S.)

**Keywords:** Spinochrome D, doxorubicin, cardioprotective effect

## Abstract

Doxorubicin, an anthracycline from *Streptomyces peucetius*, exhibits antitumor activity against various cancers. However, doxorubicin is cardiotoxic at cumulative doses, causing increases in intracellular reactive oxygen species in the heart. Spinochrome D (SpD) has a structure of 2,3,5,6,8-pentahydroxy-1,4-naphthoquinone and is a structural analogue of well-known sea urchin pigment echinochrome A. We previously reported that echinochrome A is cardioprotective against doxorubicin toxicity. In the present study, we assessed the cardioprotective effects of SpD against doxorubicin and determined the underlying mechanism. ^1^H-NMR-based metabolomics and mass spectrometry-based proteomics were utilized to characterize the metabolites and proteins induced by SpD in a human cardiomyocyte cell line (AC16) and human breast cancer cell line (MCF-7). Multivariate analyses identified 12 discriminating metabolites (variable importance in projection > 1.0) and 1814 proteins from SpD-treated AC16 cells. Proteomics and metabolomics analyses showed that glutathione metabolism was significantly influenced by SpD treatment in AC16 cells. SpD treatment increased ATP production and the oxygen consumption rate in D-galactose-treated AC16 cells. SpD protected AC16 cells from doxorubicin cytotoxicity, but it did not affect the anticancer properties. With SpD treatment, the mitochondrial membrane potential and mitochondrial calcium localization were significantly different between cardiomyocytes and cancer cell lines. Our findings suggest that SpD could be cardioprotective against the cytotoxicity of doxorubicin.

## 1. Introduction

Echinochrome A has a chemical structure of 6-ethyl-2,3,5,7,8-pentahydroxy-1,4-naphthoquinone, which exhibits cardioprotective activity and reduces the myocardial ischemia/reperfusion injury via its antioxidant effect and enhancement of mitochondria biogenesis [1,2,3]. Echinochrome A has a number of structural analogues and together they comprise the class of spinochrome pigments of sea urchins. Biological effects of spinochromes were investigated mainly on crude extracts [4] and there is not so much information on the activity of individual pigments, particularly regarding cardioprotective ability. Spinochrome D (SpD) is one of six main spinochromes and it is biosynthesized by many sea urchin species (Figure 1A) [5]. SpD is a side product of echinochrome A isolation from the flat sea urchin *Scaphechinus mirabilis*, which is utilized for the preparation of the active substance of the antioxidant and the cardioprotective drug *Histochrome*^®^ [6]. Nevertheless, the content of SpD in sea urchins is usually pretty low (0.001–0.003% of dry weight), but recently by Balaneva et al. was developed a simple and effective synthesis scheme with the yield of SpD in 58% [7]. SpD might be assumed to inherit the cardioprotective ability of echinochrome A, but the detailed mechanism has been unknown.

Doxorubicin is an anthracycline that was firstly extracted from *Streptomyces peucetius* and it has been routinely used for the treatment of several cancers, including breast, lung, gastric, ovarian, non-Hodgkin’s, and Hodgkin’s lymphoma [8,9]. There are several proposed anti-cancer mechanisms of doxorubicin, including intercalation into DNA and generation of reactive oxygen species (ROS). The inhibition of topoisomerase II has been known to induce apoptosis in cells [10,11,12].

Unfortunately, cumulative dose of doxorubicin over 550 mg/m^2^ body surface area has been known to develop cardiomyopathy [13,14]. The exact mechanism of cardiomyopathy is still controversial but iron-related free radical formation and mitochondrial disruption have been considered to be the main causes [15]. There have been trials to overcome the doxorubicin’s cardiac toxicity by reducing its oxygen radical formation to achieve important accomplishment [16,17,18].

In the present study, we demonstrated that SpD has a cardio-protective effect against the cardiac toxicity of doxorubicin without interfering the cytotoxicity to cancer cell lines. We used an integration of ^1^H-NMR based metabolomics and mass-spectrometry based proteomics to specify molecular pathways that are affected by SpD treatment in human cardiomyocytes. We also measured changes of the mitochondrial membrane potential and mitochondrial calcium in SpD treated human cardiomyocytes.

## 2. Results

### 2.1. SpD Protected AC16 Cells against Doxorubicin Cytotoxicity

Human cardiomyocyte AC16 cells were treated with SpD at 1–200 μM (Figure 1B). We found no harmful effects on cell viability. Identical concentrations of SpD were tested on rat cardiomyocyte H9c2 cells and also they showed no effect on cell viability (Appendix A). The cell viability of AC16 cells decreased after 0.1 μM doxorubicin treatment for 24 h, and an exposure duration of 48 h resulted in cell death for the majority of cells (data not shown). However, adding 10 μM SpD treatment in addition to doxorubicin for 24 h led to 80–90% AC16 cell survival (Figure 1C).

### 2.2. Liquid Chromatography–Mass Spectrometry-MS (LC-MS/MS)-Based Proteomics Analyses of SpD/Doxorubicin-Treated AC16 Cells

We performed LC-MS/MS analysis of cell lysates from SpD-treated and -untreated AC16 cells with or without doxorubicin for 24 h. The detected proteins are shown in Venn diagrams (Figure 2A,B) and full lists (Appendix A). We found networks of proteins forming clusters that are centered on “mitochondria” (Figure 2C and Appendix A). The affected pathways based on the Kyoto Encyclopedia of Genes and Genomes (KEGG) database listed proteins participating in gap junction, focal adhesion, aminoacyl-tRNA biosynthesis, and glutathione metabolism (Figure 2D).

### 2.3. ^1^H-NMR Mediated Metabolomics Analysis of SpD-Treated AC16 Cells

To identify metabolite alterations that are induced by SpD, we used ^1^H-NMR spectroscopy to characterize 30 mg of AC16 cells incubated with SpD (10 μM) for 24 h. Metabolic profiling (Chenomx, Edmonton, AB, Canada) was used to identify 32 metabolites in SpD-treated AC16 cells and untreated control cells (Table 1).

After normalization of the data (Figure 3), univariate and multivariate statistical analyses were used to comprehensively evaluate the effects of SpD on AC16 cells.

Univariate volcano plots of log2(FC) > 1.2 (*p* < 0.05) metabolites showed that the levels of sn-glycero-3-phosphocholine (GPC), glutathione, myo-inositol, taurine, and O-phosphocholine were increased, while the levels of acetate and glutamine were decreased, by SpD (Figure 4).

Multivariate analysis is used to determine the relative differences in two or more systems that are large and complex. Therefore, as shown in Figure 5A, we performed principal component analysis (PCA) of metabolites from SpD-treated AC16 cells. The aim of PCA is to reduce the dimensionality of original data within the preservation of the variance.

To calculate variable importance in projection (VIP) scores of metabolites, we performed partial least-squares projections for latent structures-discriminant analysis (PLS-DA). Metabolites with VIP scores larger than 1.0 were considered as important (Appendix A). To confirm the “goodness” of the model and the predictive quality, we tested orthogonal partial least-squares projections to latent structures-discriminant analyses (OPLS-DA) on data from SpD-treated AC16 cells and control cells (Appendix A). In PCA, the SpD-treated group and control group revealed class differences showing 95% confidence regions separating each other. We extended the supervised PLS regression using orthogonal signal correction filters after selecting VIP > 1.0 metabolites. The metabolites from the SpD-treated AC16 cells significantly differed from the control cell group in the OPLS-DA model. The R^2^Y model quality parameter was 0.937, demonstrating that the OPLS-DA model was robust (R^2^Y value near 1.0), and the Q^2^ parameter was 0.597, showing that the model was predictive (Q^2^ > 0.5) (Appendix A). The loading plot of OPLS-DA is shown in Figure 5C. The heat-map analysis of VIP > 1.0 metabolites was represented with logarithmic fold changes (Figure 5B). In comparison with the control group, the most increased and decreased metabolites with SpD treatment were GPC and acetate, respectively.

To interpret metabolic changes, correlation networks were generated according to Pearson’s correlation coefficients (|r| > 0.9) between metabolites, in a pair-wise fashion. In untreated controls, acetate shared 28 correlations and GPC shared one correlation with other metabolites. Upon SpD treatment, the number of correlations with acetate decreased to eight metabolites, while the number of metabolites correlating with GPC increased to 11 (Figure 6A,B). Pathway enrichment analyses showed that various metabolic processes, including inositol phosphate metabolism, glycerolipid metabolism, and glutathione metabolism, were involved in the SpD treatment effects (Figure 6C and Appendix A).

### 2.4. Glutathione Metabolism in AC16 Cells Was Significantly Influenced by SpD Treatment

The integration of the metabolomic and proteomic data was carried out by Integrated Molecular Pathway Level Analysis (IMPaLa) to identify significantly influenced pathways from SpD-treated AC16 cells. The KEGG database showed that VIP > 1.0 metabolites were related to nine over-represented pathways, including glutathione metabolism, protein digestion and absorption, gap junction, and sulfur metabolism (Table 2). The related genes and metabolites are presented in Appendix A. The directions of changes in the metabolite-specified proteins are listed in Table 3.

### 2.5. SpD Functioned as an Antioxidant in AC16 Cells

To test the antioxidant ability of SpD, oxidative stress was induced in AC16 cells using H_2_O_2_ or cobalt chloride with high glucose for 24 h. The DCF-DA fluorescence showed that SpD treatment reduced ROS in AC16 cells (Figure 7).

### 2.6. SpD Increased Mitochondrial ATP Production and Oxygen Consumption

SpD treatment increased intracellular ATP production and the oxygen consumption rate (OCR) in AC16 cells (Figure 8A,B). In addition, SpD treatment increased ATP production in H_2_O_2_-treated cells (Figure 8C). Co-treatment with doxorubicin (0.1 μM) and SpD for 24 h increased ATP production as compared to doxorubicin treatment alone (Figure 8D). In our study, SpD showed enhanced antioxidant capacity when compared with equimolar echinochrome A (Appendix A).

### 2.7. SpD Protected against Doxorubicin-Induced Mitochondrial Damage in AC16 Cells

Using tetramethylrhodamine (TMRE) and rhodamine-2 (rhod-2) staining, we compared mitochondrial membrane potential and mitochondrial Ca^2+^ in doxorubicin and SpD-treated AC16 cells (Figure 9). Doxorubicin treatment for 24 h decreased the TMRE intensity of AC16 cells in a dose-dependent manner. The doxorubicin-treated cells also showed increased cytosolic diffusion of rhod-2. We treated AC16 cells with 10 μM of SpD in the presence of 0.1–1.0 μM doxorubicin. SpD treatment attenuated the loss of mitochondrial membrane potential that is induced by doxorubicin (Figure 9A,B). SpD treatment reduced the cytosolic overload of mitochondrial Ca^2+^ in doxorubicin-treated cells (Figure 9C,D).

### 2.8. SpD Did Not Interfere with the Anticancer Effects of Doxorubicin in MCF-7 Cells

SpD did not inhibit the cytotoxic activity of doxorubicin in MCF-7 cells (Figure 10). Human breast cancer MCF-7 cells and human cervical cancer HeLa cells showed reduced cell viability after treatment with SpD above 100 μM (Figure 10A and Appendix A). When the SpD/doxorubicin co-treated cells were compared with the cells that were treated with doxorubicin alone, there were no significant differences in ROS level (Figure 10B), loss of mitochondrial membrane potential (Figure 10C,D), or Ca^2+^ overload (Figure 10E,F). Similar experiments using the HeLa human cervical cancer cell line showed the same lack of inhibition of cytotoxicity (Appendix A). Interestingly, SpD treatment inhibited cell migration in a wound healing test (Appendix A).

## 3. Discussion

There have been many efforts to overcome the cardiotoxicity of doxorubicin in cancer treatment. The IC_50_ (drug concentration required to inhibit cell growth by 50%) of doxorubicin for breast cancer cell lines has been reported to be between 1–4 μM after 24 h treatment (IC_50_ = 1 μM for MCF-10F; 4 μM for MCF-7; and, 1 μM for MDA-MB-231 cells) [19]. We used 10 μM of SpD, which induced no significant viability changes in either cardiomyocytes or cancer cells. From animal models using Histochrome^®^ (echinochrome A), 1–10 mg/kg of doses have been reported to act as antioxidant in cardiomyocytes, which approximately correspond to 3–30 μM [20]. In our study, echinochrome A and SpD showed cardioprotective activity when treated with 0.1 μM doxorubicin. In equimolar treatment, SpD showed better antioxidant activity and ATP production than echinochrome A. It might be reasonable to assume that cell viability decreases with increased doxorubicin incubation time. However, lower concentrations (<0.25 μM) of doxorubicin often show a low dose–time response relationship in cancer cells [21]. Cardiomyocytes and cancer cells have different mechanisms of doxorubicin-induced apoptosis. In cardiomyocytes, doxorubicin induces apoptosis by a H_2_O_2_-mediated mechanism, which is largely independent of p53 activation. In contrast, the p53 tumor suppressor plays an important role in doxorubicin-induced apoptosis in cancer cells [22,23,24].

Univariate analyses, such as fold change comparison, *t*-test, and volcano plot suggest overall shapes of measured data and multivariate analyses, including PCA, PLS-DA, and OPLS-DA, often reveal the latent structure of the data. When quantitatively analyzing multi-parametric metabolite responses, it is critical to specify all of the independent and dependent variables to be included [25]. In biological systems, metabolites are the end product of enzymatic and other protein activity, and therefore they are not independent from biological interactions. In our study, correlation network analysis and pathway enrichment analysis of VIP > 1.0 metabolites (e.g., GPC, acetate) showed that glycerolipid metabolism, glutathione metabolism, and pyruvate metabolism were significantly affected by SpD treatment in cardiomyocyte [26]. In addition to production during ethanol metabolism, acetate is transported into cells by proteins of the monocarboxylate transporter family or it is generated intracellularly by protein deacetylases and acetyl-CoA hydrolases [27,28]. In the cytosol, CoA is acetylated by acetyl-CoA synthetase to produce acetyl-CoA. In contrast, in mitochondria, acetyl-CoA is produced through the pyruvate dehydrogenase complex reactions. Acetyl-CoA participates in the citric acid cycle and β-oxidation of fatty acids to produce cellular energy (e.g., ATP). In addition to increased acetate consumption, SpD might increase cytosolic glycolysis and entrance of glutamate into the citric acid cycle, which could be shown by an increased lactate concentration and increased ratio of glutamate to glutamine (from 20.9 to 35.3; calculated as shown in Table 1). SpD treatment increased the accumulation of cytosolic osmolytes such as GPC, myo-inositol, and free amino acids (e.g., taurine and glycine), which are critical for the viability of cells. The integrated analysis with mass spectrometry based proteomics indicated that the glutathione metabolism of AC16 cells was most affected by SpD treatment. Since we measured the reduced form of glutathione (GSH) using Chenomx NMR Suit 7.1, the increased concentration of glutathione represents the increase of GSH _reduced glutathione_/GSSG _oxidized glutathione_ ratio. Using the luciferase mediated method, we confirmed the increase of GSH/GSSG ratio in a dose-dependent manner (Appendix A).

Based on the acquired results, we hypothesized that SpD mainly exerted its function as antioxidant in the process of protection of cardiomyocytes against the cytotoxicity of doxorubicin. Since the cytotoxicity of doxorubicin on cardiomyocytes is known to be based on ROS increase, we tested SpD activity in ROS generating environments. We tested 1 mM H_2_O_2_ concentration, which might be increased by constitutively active NADPH oxidase 4 (NOX4) [29]. In addition, we assessed the antioxidant ability of SpD in cobalt chloride and hyperglycemic condition. The hypoxia mimetic cobalt chloride and hyperglycemic concentrations of glucose (33.3 mM) are known to increase intracellular ROS [30,31,32,33,34]. Hyperglycemia induces hypoxia-induced cell death via the influx of calcium in diabetic cardiomyopathy [35,36,37]. Oxidative stress might cause cardiac mitochondrial dysfunction, leading to cell death [38,39].

Since the integrated analysis that is located the mitochondrial proteins clustered together at SpD treatment, we had to focus the mitochondrial ATP production by SpD treatment. To differentiate ATP production from the cytosol versus mitochondria, we added d-galactose (10 mM) to the culture media. By competing with glucose in the cytosol, d-galactose reduces cytosolic glycolysis, resulting in decreased cytosolic ATP production. Galactokinase produces galactose 1-phosphate from galactose, utilizing ATP. Uridine diphosphate (UDP)-galactose 4-epimerase converts UDP-glucose and galactose 1-phosphate into UDP-galactose and glucose 1-phosphate, respectively. Galactose participates in glycolysis by consuming ATP and reducing cytosolic glycolysis rates [40,41,42]. SpD treatment increased ATP production, even in 10 mM galactose media, which suggested the enhancement of mitochondrial ATP production with increased OCR.

In cardiomyocytes, doxorubicin has been known to induce oxidized state in mitochondrial redox potential to trigger mitochondrial depolarization and elevated calcium levels, which suppresses ATP production via oxidative phosphorylation.

As the mitochondrial dysfunction occurs, the cells are subjected to ATP depletion and become more dependent on ADP metabolism to compensate the ATP/ADP ratio [43]. In our study, the SpD treatment did not inhibit the anticancer activity of doxorubicin while protecting cardiomyocytes at identical concentration via increasing ATP production. Our approaches might provide some clues for the potential cardioprotective mechanisms of SpD in a combination therapy with doxorubicin. Nevertheless, further studies are still needed for evaluation of the drug.

## 4. Materials and Methods

### 4.1. Cell Culture and Treatment

Human cardiomyocyte cell line AC16 were purchased from Merch Company (SCC 109). Rat cardiomyocyte cell line H9c2 (CRL-1466), human breast cancer cell line MCF-7 (HTB-22), and human cervical cancer cell line HeLa (CCL-2) were purchased from America Tissue Type Collection (ATCC, Bethesda, Rockville, MD, USA). Cells were routinely cultured in Dulbecco’s modified Eagle’s medium (DMEM) containing 25 mmol/L glucose and l-glutamine (Sigma, St. Louis, MO, USA) and 10% fetal bovine serum (FBS) (HyClone, Logan, UT, USA). Cells were maintained at 37 °C and 5% CO_2_. For 10 mM galactose DMEM, d-galactose (Sigma, St. Louis, MO, USA) was added to the media. Spinochrome D (SpD, purity 98%) was isolated from the sea urchin *Scaphechinus mirabilis*, as described in [44], as the red powder with m.p. >320 °C and spectral characteristics, as in [45].

#### 4.1.1. Cell Viability Assay

Cells were seeded in 96-well plates at 2 × 10^4^ cells/well until adherent. Cells were treated with SpD for 24 h and then 10 μL of Cell Counting Kit-8 reagent (CCK-8, Dojindo Molecular Technologies, Kyushu, Japan) was added into each well for 20–30 min at 37 °C. The absorbance was read using a SpectraMax microplate reader (Molecular Devices, San Jose, CA, USA) at 450 nm.

#### 4.1.2. Intracellular ATP Measurement

The ATP levels that are produced by cardiomyocytes were measured by the Luciferin-Luciferase reaction, according to the manufacturer’s instructions (Cayman Chemicals, Ann Arbor, MI, USA). Cells were plated at 2 × 10^4^ cells/well in 96-well plates and incubated with test compounds at 37 °C and 5% CO_2_ for 24 h. Cells were lysed using 50 μL of lysis buffer (Triton X100; 0.1%), mixed with 50 μL of ATP measurement solution containing Luciferin-Luciferase, and then incubated at room temperature for 10–15 min. The luminescence was read using a luminometer (SpectraMax M2e; Molecular Devices, Sunnyvale, CA, USA) and expressed as relative light units. The ATP levels of the control and drug-treated samples were compared to ensure that the reading was exclusive to the ATP produced by drug-treated cardiomyocytes.

#### 4.1.3. Oxygen Consumption Ratio (OCR) Measurement

The OCR in cardiomyocytes was measured using a MitoXpress^®^ probe (Cayman Chemicals, Ann Arbor, MI, USA) according to the manufacturer’s instructions. Cells were plated at 2 × 10^4^ cells/well in 96-well plates and incubated with SpD at 37 °C and 5% CO_2_ for 24 h. After replacing the spent media with 160 μL of 10% FBS-DMEM containing SpD, 10 μL of MitoXpress^®^ Xtra Solution, and 100 μL of mineral oil were added to all wells. The plates were incubated at 37 °C and 5% CO_2_ for 10–20 min and then the fluorescence was read at 380 nm excitation/650 nm emission on a fluorometer with a delay time of 100 μs (SpectraMax M2e).

#### 4.1.4. Intracellular ROS Levels

The ROS levels were measured using 2′,7′-dichlorofluorescein diacetate (DCF-DA) (Sigma-Aldrich). Cells were plated at 2 × 10^4^ cells/well in 96-well plates and incubated with test compounds at 37 °C and 5% CO_2_ for 24 h. Subsequently, the cells were washed three times with phosphate-buffered saline (PBS) and incubated in the dark with 20 μM DCF-DA for 30 min at 37 °C. The cells were washed twice with PBS. The fluorescence was measured using a fluorometer at 485 nm excitation/535 nm emission.

#### 4.1.5. Measurement of Mitochondrial Membrane Potential

Mitochondrial membrane potential was measured using tetramethylrhodamine (TMRE; Thermo Fisher Scientific, Scotts Valley, CA, USA). Cells were incubated with 200 nM TMRE in the dark for 30 min at 37 °C and 5% CO_2_. After washing, the fluorescence was measured at 550 nm excitation/590 nm emission and cells were imaged using a fluorescence microscope (Olympus, IX71; Olympus, Tokyo, Japan).

#### 4.1.6. Measurement of Mitochondrial Calcium

Mitochondrial calcium was measured using rhod-2/AM (TMRE; Thermo Fisher Scientific). Cells were incubated with 1 μM rhod-2 for 30 min. After PBS washing, the fluorescence was measured at 552 nm excitation/581 nm emission and cells were imaged using a fluorescence microscope.

#### 4.1.7. Cell Migration Assay

Scratch wound assays were used for cancer cell mobility analyses. MCF-7 and HeLa cells were seeded into 25-well cell culture plates at a concentration of 2 × 10^4^ cells and were maintained in 10% FBS-DMEM until 70–80% confluent. The scratch was carefully made using a 20P sterile pipette tip. The remaining cellular debris was gently removed with PBS. The wounded monolayer was incubated in 10% FBS-DMEM containing 10 μM SpD for 24 h. The cell migration was observed under 4× by using a phase contrast microscope.

### 4.2. LC-MS/MS Analysis and Database Searching

The trypsinized peptides from 50 μg of SpD-treated cell supernatant were analyzed using an LTQ-Orbitrap Velos^TM^ mass spectrometer coupled with an EASY-nLC II (Thermo Fisher Scientific, Waltham, MA, USA). The Uniprot human database was used for peptide searching. The protein identification was confirmed if the normalized fold change (FC) was higher than 1.30 (upregulated) or less than 0.77 (downregulated). The confidence level (CI) was 95% based on pair-wise analyses as compared with untreated controls. Relative peptide abundance was quantified with Scaffold using the Top 3 total ion chromatogram method. The differentially expressed proteins were categorized into Gene Ontology terms, i.e., biological process, cellular component, and molecular function.

### 4.3. ^1^H-NMR Metabolomics

The high-resolution magic-angle spinning nuclear magnetic resonance (HR-MAS NMR) spectra were recorded using an Agilent 600 MHz spectrometer that was equipped with a 4 mm gHX NanoProbe (Agilent Technologies, Santa Clara, CA, USA). All spectra were acquired at 600.167 MHz. The acquisition time was 1.703 s, relaxation delay was 1 s, and a total of 128 scans was obtained. The Carr-Purcell-Meiboom-Gill (CPMG) pulse sequence was used for the suppression of water and compounds with high molecular mass. For data processing, Chenomx NMR Suit 7.1 professional with the Chenomx 600 MHz library database were used (Chenomx Inc., Edmonton, AB, Canada). The bin size for spectra was 0.001 ppm. The binning data were normalized to the total area. PCA, partial least-squares discriminant analysis (PLS-DA), and orthogonal partial least-squares discriminant analysis (OPLS-DA) were performed using SIMCA-P+ 12.0 (Umetrics, Malmö, Sweden). For visualization of VIP scores of metabolites and Metabolic Set Enrichment Analysis, web-based software MetaboAnalyst 3.0 (http://www.metaboanalyst.ca) was used.

### 4.4. Pathway Enrichment Analysis

Integrated Molecular Pathway Level Analysis (IMPaLa, http://impala.molgen.mpg.de/) was used to specify the pathways that are affected by SpD. Only VIP > 1.0 metabolites and Log2(FC) > 1.2 proteins were considered. Search Tool for the Retrieval of Interacting Genes/Proteins (STRING, https://string-db.org) and Cytoscape (downloaded at https://cytoscape.org) were used for clustering molecular networks.

## 5. Conclusions

The present study investigated the effect of SpD on doxorubicin-treated cardiomyocytes through an integration of metabolic and proteomic analyses. Univariate and multivariate analyses of ^1^H-NMR spectroscopy data identified the potentially affected metabolites and groups of proteins from SpD-treated cardiomyocytes. Based on the ^1^H-NMR data, SpD increased glutathione, which regulates intracellular ROS stress. In addition, SpD treatment increased cytosolic and mitochondrial ATP production in cardiomyocytes, which was significantly correlated with increased lactate and decreased acetate levels. Co-treatment with SpD protected doxorubicin-treated cardiomyocytes, reducing the mitochondrial damage of doxorubicin. In contrast, SpD did not inhibit the cytotoxicity of doxorubicin in cancer cells. The integrated metabolomics and proteomics data suggest the involvement of the Akt/mTOR signaling pathway by which SpD might protect cardiomyocytes (Table 3). However, further study is still needed to verify these relationships.

## Figures and Tables

**Figure 1 marinedrugs-17-00002-f001:**
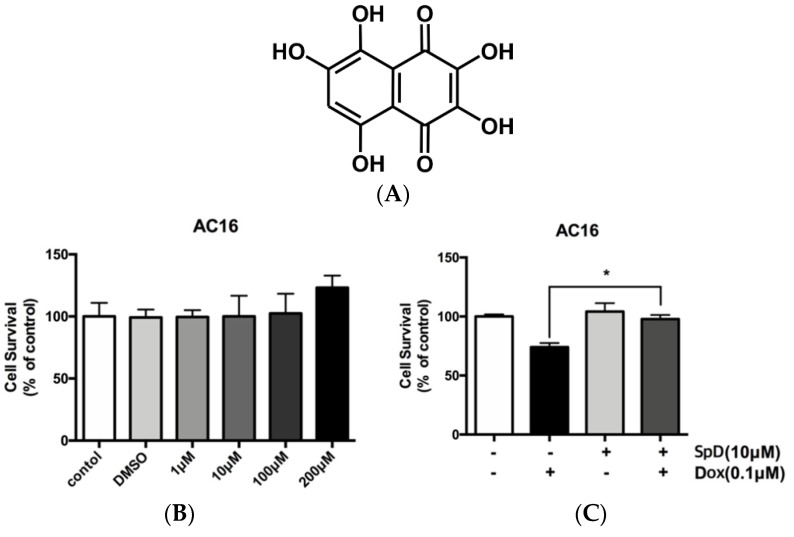
Spinochrome D (SpD) protected cardiac cells from the cytotoxicity of doxorubicin. (**A**) Chemical structure of SpD (MW, 238.15044); (**B**) AC16 human cardiomyocytes were treated with 0–200 μM SpD for 24 h and cell viability was measured using a Cell Counting Kit-8 reagent (CCK-8) assay. SpD did not affect the cell viability of cardiomyocytes. * *p* < 0.05 compared with control; (**C**) Treatment with SpD (10 μM, 24 h) attenuated the cardiotoxicity of doxorubicin (0.1 μM) in AC16 cells.

**Figure 2 marinedrugs-17-00002-f002:**
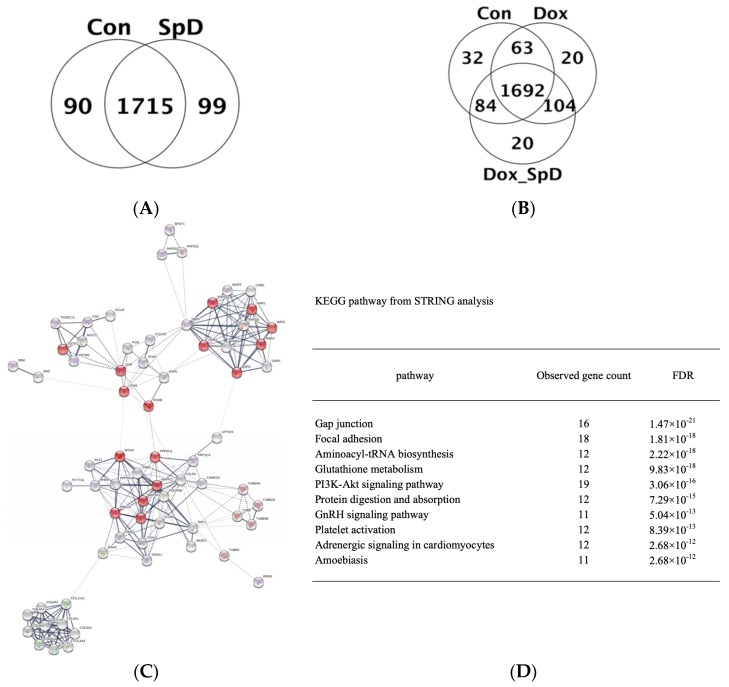
Mass spectrometry-based proteomics of SpD-treated AC16 cells. Liquid Chromatography–Mass Spectrometry-MS (LC-MS/MS) spectrometry-based proteomics detected proteins from SpD (10 μM, 24 h) (**A**) and SpD/doxorubicin (**B**) treated AC16 cells; (**C**) Search Tool for the Retrieval of Interacting Genes/Proteins (STRING) analysis showed that altered metabolic proteins clustered around “mitochondria” which are represented as red colored nodes. All filled nodes represent the 3D structures of proteins are known; and, (**D**) The top 10 influenced metabolic pathways are shown from the STRING analysis (Kyoto Encyclopedia of Genes and Genomes (KEGG) database).

**Figure 3 marinedrugs-17-00002-f003:**
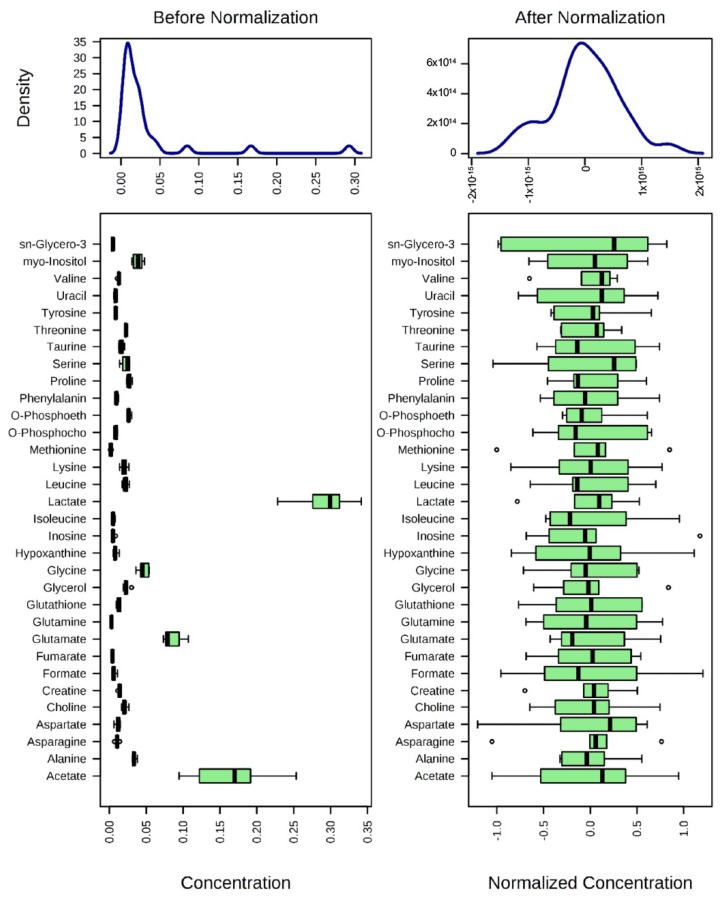
Normalization of ^1^H-NMR aquired metabolite concentrations. The concentrations of metabolites were normalized by log-transformation followed by Pareto scaling (mean-centered and divided by the square root of the standard deviation of each variable). Changes of metabolites are represented as ratios of control metabolites.

**Figure 4 marinedrugs-17-00002-f004:**
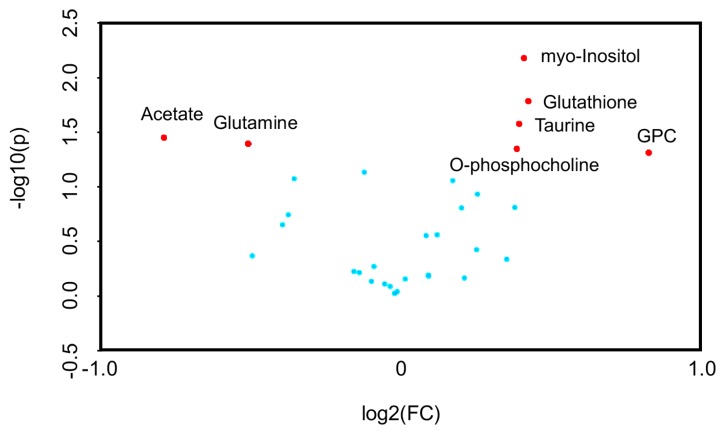
Volcano plots for SpD-induced metabolic changes compared with controls (*n* = 3). Metabolites are considered significant if log2(fold change) > 1.2. The *p*-value threshold was 0.05. The significantly changed metabolites included acetate, glutamine, myo-inositol, glutathione, taurine, O-phosphocholine, and sn-Glycero-3-phosphocholine (GPC).

**Figure 5 marinedrugs-17-00002-f005:**
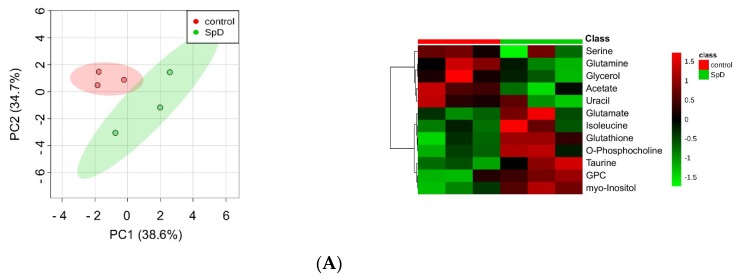
^1^H-NMR metabolomics for SpD-treated AC16 cells. (**A**) Principal component analysis (PCA) indicated that metabolites from the SpD-treated (10 μM, 24 h) group were significantly different from those in the control group; (**B**) Heat-map analysis of metabolites with variable importance in projection (VIP) score > 1.0. The logarithmic fold changes are shown below. GPC, sn-glycero-3-phosphocholine; (**C**) The loading plots from orthogonal partial least-squares discriminant analysis (OPLS-DA) for SpD metabolites compared with the control group.

**Figure 6 marinedrugs-17-00002-f006:**
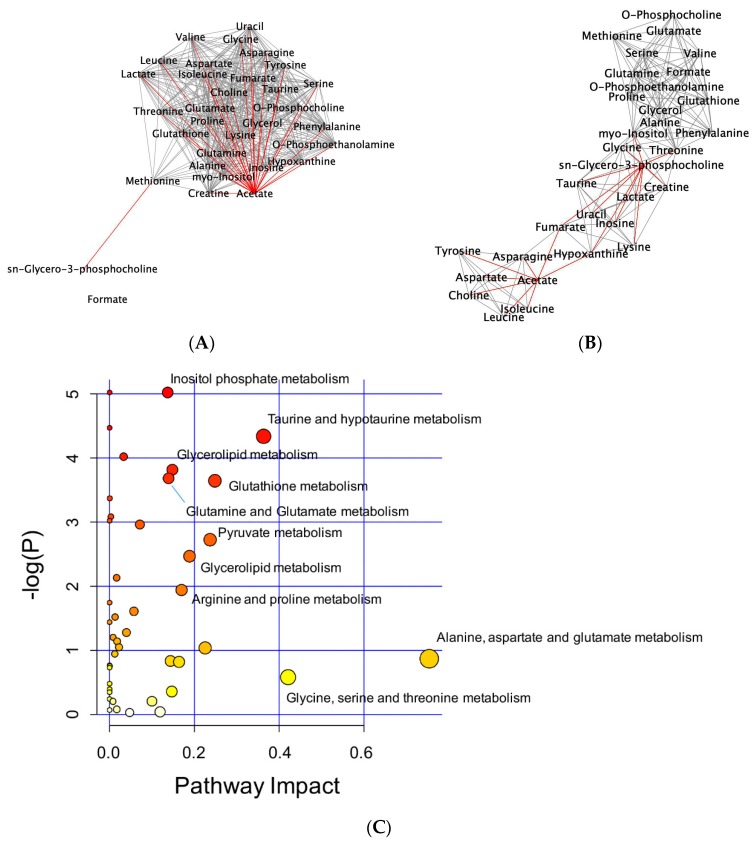
Network analysis of metabolites altered by SpD (10 μM, 24 h) treatment of AC16 cells. Networks of metabolites according to their Pearson’s correlation coefficients were drawn using Cytoscape program. The networks with significantly increased GPC and decreased acetate are marked as red lines. (**A**) control and (**B**) SpD-treated cell metabolites; and, (**C**) Pathway impact analysis shows the most affected metabolic pathways affected by SpD. Varying colors from yellow to red represent metabolites’ significance in the data.

**Figure 7 marinedrugs-17-00002-f007:**
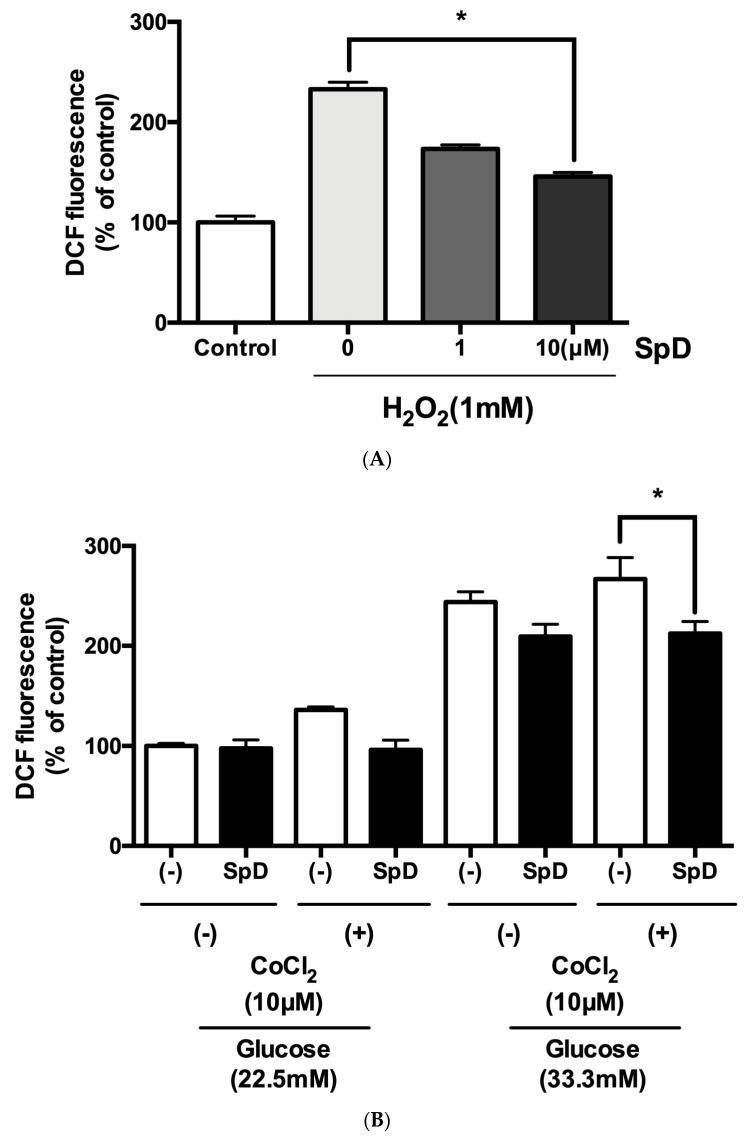
SpD showed antioxidant activity in AC16 cells treated with H_2_O_2_ or cobalt chloride and hyperglycemic stress. (**A**) Reactive oxygen species generation was induced in AC16 cells using 1 mM H_2_O_2_ and SpD (0–10 μM) was co-treated for 24 h. * *p* < 0.05 compared with 1mM H_2_O_2_ group without SpD; (**B**) Cobalt chloride (a hypoxia-mimetic agent) and hyperglycemia (33.3 mM glucose in media) were applied to AC16 cells. The cells were incubated with ′,7′-dichlorofluorescein diacetate (DCF-DA) (20 μM) for 20 min at 37 °C and the intensity of fluorescence was measured at 485 nm. * *p* < 0.05 compared with no SpD treated group.

**Figure 8 marinedrugs-17-00002-f008:**
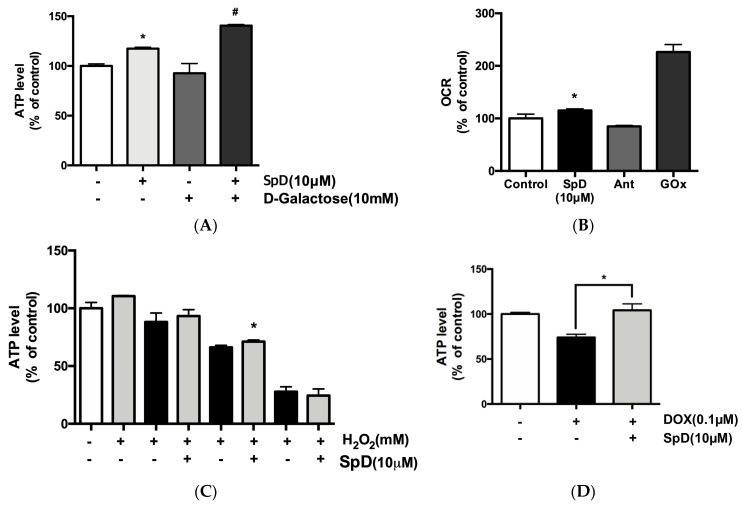
SpD caused increased ATP production and oxygen consumption rate (OCR) in AC16 cells. (**A**) SpD (10 μM) increased ATP production in AC16 cells. D-galactose (10 mM) was added to reduce cytosolic glycolytic ATP production. By changing energy metabolism in cardiomyocytes by replacing glucose with galactose, high concentrations of galactose could prevent ATP production except that of mitochondria by oxidative phosphorylation (OXPHOS); (**B**) SpD (10 μM) increased OCR. Antimycin A (Ant, 1 μM, a Complex III inhibitor) was used as a cell-based negative control and glucose oxidase (GOx, 1 mg/mL) was used as a cell-free positive control. (**C**) SpD increased ATP levels under H_2_O_2_ induced oxidative stress. (**D**) SpD increased ATP production in the presence of doxorubicin (0.1 μM). * *p* < 0.05 compared with untreated controls, # *p* < 0.05 compared with the D-galactose-treated group.

**Figure 9 marinedrugs-17-00002-f009:**
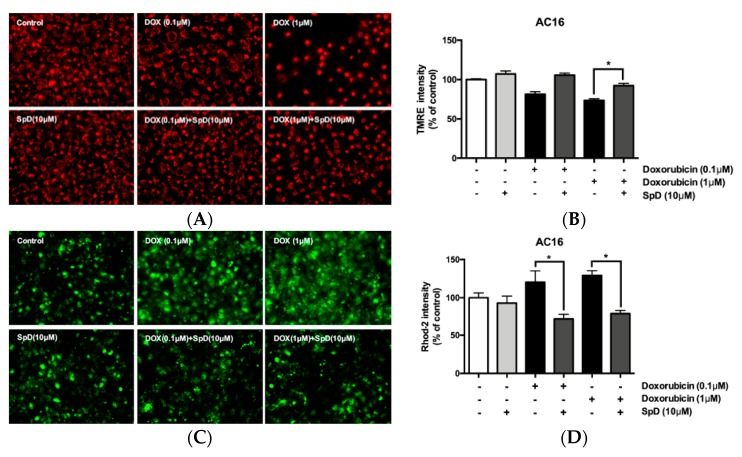
SpD attenuated doxorubicin-induced mitochondrial membrane potential and mitochondrial calcium changes in AC16 cells. (**A**) Mitochondrial membrane potential (Δψ_m_) in AC16 cells was indicated by TMRE fluorescence. The cells were treated with doxorubicin (0.1–1.0 μM) with/without SpD (10 μM); (**B**) The intensity of tetramethylrhodamine (TMRE) staining was measured using fluorometry at 550_Ex_/590_Em_ nm. Doxorubicin decreased mitochondrial membrane potential in a dose-dependent manner. Co-treatment with SpD attenuated the membrane potential loss; (**C**) Mitochondrial calcium was localized using rhod-2, a selective indicator for mitochondrial Ca^2+^; and, (**D**) The intensity of rhod-2 was measured at 552_Ex_/581_Em_ nm. Doxorubicin induced diffusion of mitochondrial Ca^2+^ to the cytosolic space but SpD co-treatment attenuated the Ca^2+^ diffusion. * *p* < 0.05 compared with the doxorubicin-treated group.

**Figure 10 marinedrugs-17-00002-f010:**
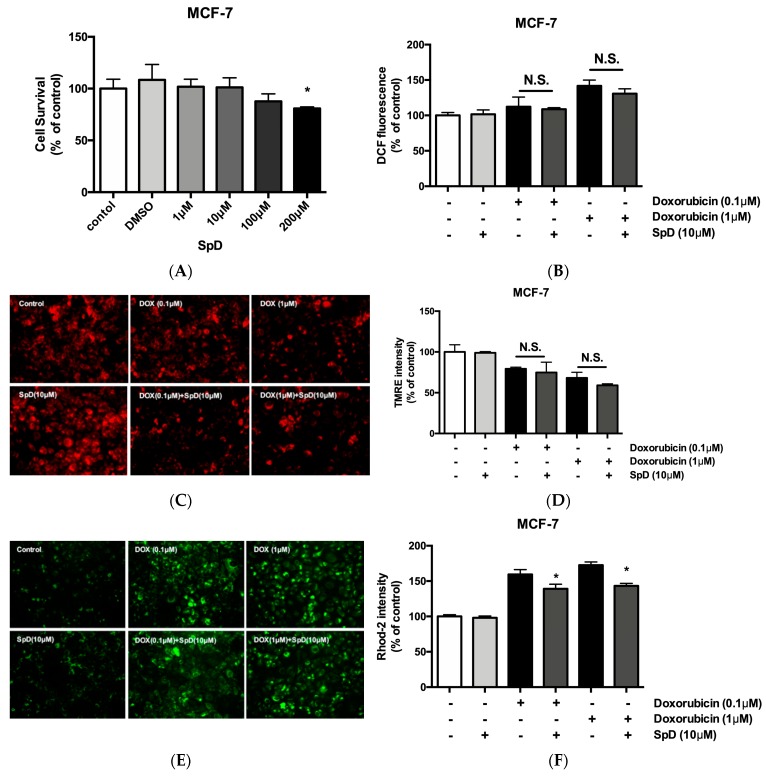
SpD did not inhibit the cytotoxicity of doxorubicin. (**A**) SpD induced MCF-7 cell death at 100–200 μM. * *p* < 0.05 compared with the untreated control; (**B**) The co-treatment of SpD and doxorubicin showed similar levels of ROS compared with the doxorubicin alone group; (**C**) Mitochondrial membrane potential was visualized using TMRE staining in SpD/doxorubicin-treated MCF-7 cells; (**D**) The TMRE intensity measured by fluorometry is shown; (**E**) Mitochondrial Ca^2+^ level is indicated using rhod-2 dye in SpD/doxorubicin-treated MCF-7 cells; (**F**) The rhod-2 intensity was measured by fluorometry. * *p* < 0.05 compared with doxorubicin treated groups without SpD. ROS = reactive oxygen species.

**Table 1 marinedrugs-17-00002-t001:** Identified metabolites and their corresponding concentrations (mM; mean, standard deviation), as determined by Chenomx NMR Suit 7.1^®^ peak fitting of individual ^1^H-NMR spectra (600 MHz) for SpD (10 μM) treated AC16 cells (30 mg, *n* = 3).

Metabolite	Control		SpD	
	Mean (mM)	S.D.	Mean (mM)	S.D.
Acetate	8.218	5.815	4.820	1.541
Alanine	1.236	0.665	1.341	0.315
Asparagine	0.388	0.194	0.397	0.095
Aspartate	0.412	0.139	0.428	0.138
Choline	0.751	0.437	0.831	0.111
Creatine	0.463	0.190	0.557	0.051
Formate	0.230	0.034	0.197	0.048
Fumarate	0.143	0.047	0.164	0.043
Glutamate	2.877	1.640	3.622	0.791
Glutamine	0.129	0.048	0.101	0.006
Glutathione	0.383	0.158	0.572	0.081
Glycerol	0.912	0.405	0.788	0.088
Glycine	1.650	0.952	1.900	0.656
Hypoxanthine	0.285	0.222	0.349	0.089
Inosine	0.169	0.068	0.209	0.056
Isoleucine	0.169	0.080	0.235	0.031
Lactate	9.551	3.793	12.374	2.715
Leucine	0.829	0.536	0.824	0.073
Lysine	0.783	0.489	0.719	0.124
Methionine	0.064	0.017	0.090	0.029
O-Phosphocholine	0.274	0.124	0.392	0.066
O-Phosphoethanolamine	0.939	0.468	1.068	0.091
Phenylalanine	0.376	0.200	0.364	0.018
Proline	0.907	0.435	1.114	0.202
Serine	0.948	0.536	0.790	0.352
Taurine	0.505	0.270	0.723	0.198
Threonine	0.853	0.434	0.846	0.152
Tyrosine	0.323	0.168	0.329	0.004
Uracil	0.363	0.243	0.279	0.021
Valine	0.479	0.272	0.474	0.073
myo-Inositol	1.198	0.554	1.750	0.352
sn-Glycero-3-phosphocholine (GPC)	0.123	0.058	0.245	0.068

**Table 2 marinedrugs-17-00002-t002:** Pathways determined from integration of metabolomic and proteomic data of SpD treated AC16 cells. Identified proteins and metabolites were analyzed using Integrated Molecular Pathway Level Analysis (IMPaLA) for pathway enrichment.

Pathway	No. of Overlapping Genes	No. of Genes in Pathway	No. of Overlapping Metabolites	No. of Metabolites in Pathway	*p*-Value	*q*-Value
Glutathione metabolism	12	54 (54)	2	38 (38)	8 × 10^−6^	4 × 10^−4^
Protein digestion and absorption	12	90 (90)	3	47 (47)	3 × 10^−5^	9 × 10^−4^
Gap junction	16	88 (88)	1	11 (11)	7 × 10^−5^	0.002
Sulfur metabolism	4	10 (10)	2	33 (33)	8 × 10^−5^	0.002
Aminoacyl-tRNA biosynthesis	12	66 (66)	2	52 (52)	9 × 10^−5^	0.002
Choline metabolism in cancer	12	99 (99)	2	11 (11)	1 × 10^−4^	0.002
Central carbon metabolism in cancer	11	65 (65)	2	37 (37)	1 × 10^−4^	0.003
Long-term depression	9	60 (60)	1	9 (9)	0.003	0.033
Long-term potentiation	9	67 (67)	1	7 (7)	0.004	0.043

**Table 3 marinedrugs-17-00002-t003:** Direction of log (FC) of metabolite-specified genes/proteins in SpD/Dox treated AC16.

Pathway	Gene	Direction of Log(FC) vs. Control	Protein
SpD	Dox	Dox and SpD
Glutathione metabolism	TXNDC12	DOWN	DOWN	UP	Thioredoxin domain-containing protein 12
Protein digestion and absorption	COL4A1	DOWN	DOWN	UP	Collagen alpha-1(IV) chain
	COL5A2	DOWN	DOWN	UP	Collagen alpha-2(V) chain
Gap junction	MAPK3	DOWN	UP	DOWN	Mitogen-activated protein kinase
	MAP2K1	UP	DOWN	UP	Dual specificity mitogen-activated protein kinase kinase 1
Sulfur metabolism	BPNT1	DOWN	DOWN	UP	3′(2′),5′-bisphosphate nucleotidase 1
Aminoacyl-tRNA biosynthesis	FARSB	DOWN	DOWN	UP	Phenylalanine--tRNA ligase beta subunit
	TARS	UP	UP	DOWN	Threonine--tRNA ligase, cytoplasmic
Choline metabolism in cancer	MAP2K1	UP	DOWN	UP	Dual specificity mitogen-activated protein kinase kinase 1
	AKT2	DOWN	UP	UP	RAC-beta serine/threonine-protein kinase
	RAC1	DOWN	DOWN	UP	Isoform B of Ras-related C3 botulinum toxin substrate 1
	RHEB	DOWN	UP	UP	GTP-binding protein Rheb
	MAPK3	DOWN	UP	DOWN	Mitogen-activated protein kinase
Central carbon metabolism in cancer	MAP2K1	UP	DOWN	UP	Dual specificity mitogen-activated protein kinase kinase 1
	PDHB	UP	DOWN	DOWN	Isoform 2 of Pyruvate dehydrogenase E1 component subunit beta, mitochondrial
	MTOR	UP	DOWN	UP	Serine/threonine-protein kinase mTOR
	AKT2	DOWN	UP	UP	RAC-beta serine/threonine-protein kinase
	MAPK3	DOWN	UP	DOWN	Mitogen-activated protein kinase
Long-term depression	MAP2K1	UP	DOWN	UP	Dual specificity mitogen-activated protein kinase kinase 1
	MAPK3	DOWN	UP	DOWN	Mitogen-activated protein kinase
Long-term potentiation	MAP2K1	UP	DOWN	UP	Dual specificity mitogen-activated protein kinase kinase 1
	CAMK2D	DOWN	UP	UP

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
