# Peer review of "Spinochrome D Attenuates Doxorubicin-Induced Cardiomyocyte Death via Improving Glutathione Metabolism and Attenuating Oxidative Stress"

_marinedrugs, 2018, doi:10.3390/md17010002_

Round 1
Reviewer 1 Report
As you described in abstract, the authors previously reported the protective effects of Echinochrome A on cardiotoxicity in cardiomyocytes in vitro. Because this present research further develops your previous research, comparison of both is important for understanding Spinochrome D derived from Echinochrome A. Therefore, the authors should summarize your reports previously published in the section of introduction. Then, it is necessary to discuss the comparison of the effects of both drugs in the section of discussion.
In this study, the authors employed breast and cervical cancer cell lines as well as cardiomyocyte. The reason for using cancer cells to elucidate cardiotoxicity is not clear. I could not find AC16 in ATCC catalog. If the source of the cell is unclear, the results will doubt. The authors must confirm the source of AC16 used in this research.
Although Spinochrome D is effective at 200 μM (Fig.1b), relatively low concentration of Spinochrome D is used in examination of protective effect (Fig.1c). How were the concentrations of Spinochrome D selected in Figures? It is necessary to clarify the concentrations of other drugs as well. References should be provided showing that these concentrations are within pharmacological or physiological ranges.
It is difficult to understand the relevance between the pathway analysis and the results of in vitro. The author should clearly describe their relevance. Separating the results and discussion will make the results easier to understand.
Author Response
Jin Han

National Research Laboratory for Mitochondrial Signaling, Department of Physiology, College of Medicine, Cardiovascular and Metabolic Disease Center (CMDC),

Inje University,

Busan 614-735

e-mail: phyhanj@gmail.com
Dear Editor-in-Chief:
We resubmit to you a revised version with excitement. Thank you for giving us an opportunity to revise our manuscript marinedrugs-406270, "Spinochrome D attenuates doxorubicin-induced cardiomyocyte death via improving glutathione metabolism and attenuating oxidative stress". We appreciate the time and details provided by each reviewer and by you. We have incorporated the suggested changes into the manuscript to the best of our ability and the manuscript has certainly benefited from these insightful revision suggestions.
We have responded specifically to each suggestion as follows.
Reviewer 1
Review Report Form
Comments and Suggestions for Authors
As you described in abstract, the authors previously reported the protective effects of Echinochrome A on cardiotoxicity in cardiomyocytes in vitro. Because this present research further develops your previous research, comparison of both is important for understanding Spinochrome D derived from Echinochrome A.
Therefore, the authors should summarize your reports previously published in the section of introduction.
-> We summarized our reports about echinochrome A in the section of introduction as in line 35-37; " Echinochrome A has a chemical structure of 6-ethyl-2,3,5,7,8-pentahydroxy-1,4-naphthoquinone which exhibits cardioprotective activity and reduces the myocardial ischemia/reperfusion injury [1-3]"
Then, it is necessary to discuss the comparison of the effects of both drugs in the section of discussion.
-> We added the comparison of the effects of both drugs in the section of discussion as in line 276-278; " In our study, echinochrome A and SpD showed cardioprotective activity when treatred with 0.1 μM doxorubicin. In equimolar treatment, SpD showed better antioxicant activity and ATP production than echinochrome A."
In this study, the authors employed breast and cervical cancer cell lines as well as cardiomyocyte.
The reason for using cancer cells to elucidate cardiotoxicity is not clear.
-> Due to the ambiguity, we deleted the contents using cancer cells in the cardiotoxicity section.
I could not find AC16 in ATCC catalog.
-> Corrected as in line 331; " Human cardiomyocyte cell line AC16 were purchased from Merch Company (SCC 109)."
If the source of the cell is unclear, the results will doubt. The authors must confirm the source of AC16 used in this research.
Although Spinochrome D is effective at 200 μM (Fig.1b), relatively low concentration of Spinochrome D is used in examination of protective effect (Fig.1c). How were the concentrations of Spinochrome D selected in Figures? It is necessary to clarify the concentrations of other drugs as well.
References should be provided showing that these concentrations are within pharmacological or physiological ranges.
-> We added the reference about the used concentration as in line 273-276; " We used 10 μM of SpD, which induced no significant viability changes in either cardiomyocytes or cancer cells. From animal models using Histochrome® (echinochrome A), 1-10mg/kg of doses have been reported to act as antioxidant in cardiomyocytes, which approximately correspond to 3-30 μM [20]."
It is difficult to understand the relevance between the pathway analysis and the results of in vitro.
-> We added the required content as in line 306-309; " Therefore we hypothesized that SpD mainly exerted its function as antioxidant in the process of protection of cardiomyocytes against the cytotoxicity of doxorubicin. Since the cytotoxicity of doxorubicin on cardiomyocytes is known to be based on ROS increase, we tested SpD activity in ROS generating environments."
The author should clearly describe their relevance. Separating the results and discussion will make the results easier to understand.
-> The results and discussion sections were separated as the reviewer suggested.
Reviewer 2 Report
In this article, Chang Shin Yoon et al. assessed the cardioprotective effects of spinochrome D (SpD) against cardiotoxicity induced by doxorubicin in human cardiomyocyte cell line (AC16) and in human breast cancer cell line MCF-7. Using high level instrumentation including NMR- based metabolomics and mass specs and methodologies have provided excellent and strong evidence that (1) SpD influences glutathione metabolism, increased ATP generation, and oxygen consumption in the normal AC16 cell line. Furthermore, they have provided strong evidence that SpD significantly protected AC16 cells from doxorubicin-induced cytotoxicity without affecting the anti-cancer properties of doxorubicin. They also provided strong evidence that SpD induces differential effects in mitochondria membrane potential and mitochondrial calcium localization in normal and cancer cells. From these results, they strongly concluded that SpD is able to terminate doxorubicin-induced cardiomyocyte death by attenuating oxidative stress while improving glutathione metabolism. In addition to these very interesting metabolic changes, the authors also identified other metabolites, which are affected by the mechanisms of SpD in the presence or absence of doxorubicin. Lastly, the author employed high powered statistical analysis include multivariate analysis for identification of at least 12 metabolites and over 1800 proteins from the SpD treated cells. The data presentation is satisfactory. In addition, the data analysis and conclusions drawn from the outcomes of the study were supportive. The manuscript is well written with very extensive results presentation and discussions though the conclusion section is relatively brief. The literature citation section is adequate.
Areas for revision:
Under the results and discussions on page 3 of the article, the authors made references to figures 10A and 7A when in fact they had not presented data in figures 2, 3 4 etc. That seem to be out of sequence with respect to how they selected to present the data in the figures. Since they wanted to discuss the results in figures 10A and 7A earlier in the manuscript, those figures should be moved ahead of figures 2 and 3. Therefore, I am recommending rearranging and renumbering the figures so that figure numbers correlate with their presentations sequence in the order in which they are presented and discussed in the article.
Since they studied the effects of SpD on glutathione metabolism, I assume they measured GSH and GSSG levels. If so, they should comment on the changes in the GSH/GSSG ratio without and without SpD.
They should comment on whether the changes in mitochondrial membrane potential correlates well with the changes in ATP synthesis; better still the ATP/ADP ratio
Author Response
Jin Han

National Research Laboratory for Mitochondrial Signaling, Department of Physiology, College of Medicine, Cardiovascular and Metabolic Disease Center (CMDC),

Inje University,

Busan 614-735

e-mail: phyhanj@gmail.com
Dear Editor-in-Chief:
We resubmit to you a revised version with excitement. Thank you for giving us an opportunity to revise our manuscript marinedrugs-406270, "Spinochrome D attenuates doxorubicin-induced cardiomyocyte death via improving glutathione metabolism and attenuating oxidative stress". We appreciate the time and details provided by each reviewer and by you. We have incorporated the suggested changes into the manuscript to the best of our ability and the manuscript has certainly benefited from these insightful revision suggestions.
We have responded specifically to each suggestion as follows.
Reviewer 2
Review Report Form
Comments and Suggestions for Authors
In this article, Chang Shin Yoon et al. assessed the cardioprotective effects of spinochrome D (SpD) against cardiotoxicity induced by doxorubicin in human cardiomyocyte cell line (AC16) and in human breast cancer cell line MCF-7. Using high level instrumentation including NMR- based metabolomics and mass specs and methodologies have provided excellent and strong evidence that (1) SpD influences glutathione metabolism, increased ATP generation, and oxygen consumption in the normal AC16 cell line. Furthermore, they have provided strong evidence that SpD significantly protected AC16 cells from doxorubicin-induced cytotoxicity without affecting the anti-cancer properties of doxorubicin. They also provided strong evidence that SpD induces differential effects in mitochondria membrane potential and mitochondrial calcium localization in normal and cancer cells. From these results, they strongly concluded that SpD is able to terminate doxorubicin-induced cardiomyocyte death by attenuating oxidative stress while improving glutathione metabolism. In addition to these very interesting metabolic changes, the authors also identified other metabolites, which are affected by the mechanisms of SpD in the presence or absence of doxorubicin. Lastly, the author employed high powered statistical analysis include multivariate analysis for identification of at least 12 metabolites and over 1800 proteins from the SpD treated cells. The data presentation is satisfactory. In addition, the data analysis and conclusions drawn from the outcomes of the study were supportive. The manuscript is well written with very extensive results presentation and discussions though the conclusion section is relatively brief. The literature citation section is adequate.
Areas for revision:
Under the results and discussions on page 3 of the article, the authors made references to figures 10A and 7A when in fact they had not presented data in figures 2, 3 4 etc. That seem to be out of sequence with respect to how they selected to present the data in the figures. Since they wanted to discuss the results in figures 10A and 7A earlier in the manuscript, those figures should be moved ahead of figures 2 and 3. Therefore, I am recommending rearranging and renumbering the figures so that figure numbers correlate with their presentations sequence in the order in which they are presented and discussed in the article.
-> The figures and content were rearranged and the explanation about figure 10A and 7A was moved to section 2.8.
Since they studied the effects of SpD on glutathione metabolism, I assume they measured GSH and GSSG levels. If so, they should comment on the changes in the GSH/GSSG ratio without and without SpD.
-> Since 1H-NMR measures the reduced form of glutathione(GSH), the increased glutathione level represents the increase of reduced form of glutathione. To confirm this, we measured GSH/GSSG ratio using luciferase based method from SpD treated AC16 cells. The GSH/GSSG results were added as Figure S9. In manuscript we added the content as in line 305-308;" Since we measured the reduced form of glutathione (GSH) using Chenomx NMR Suit 7.1, the increased concentration of glutathione represents the increase of GSHreduced glutathione/GSSGoxidized glutathione ratio. Using luciferase mediated method, we confirmed the increase of GSH/GSSG ratio by a dose-dependent manner (Figure S9)."
They should comment on whether the changes in mitochondrial membrane potential correlates well with the changes in ATP synthesis; better still the ATP/ADP ratio
-> We commented the correlation of loss of mitochondrial membrane potential with the change in ATP production and calcium overload as in line 329-334;" In cardiomyocytes, doxorubicin has been known to induce oxidized state in mitochondrial redox potential to trigger mitochondrial deplorization and elevated calcium levels. As the mitochondrial dysfunction occrurs, the cells are subjected to ATP depletion and become more dependent on ADP metabolism to compensate ATP/ADP ratio [43]. In our study, the SpD treatment did not inhibit the anticancer activity of doxorubicin while protecting cardiomyocytes at identical concentration via increasing ATP production."
Round 2
Reviewer 1 Report
There is no more comment regarding this manuscript.